# Increased Mortality Risk at Septic Condition in Inflammatory Skin Disorders and the Effect of High-Fat Diet Consumption

**DOI:** 10.3390/ijms25010478

**Published:** 2023-12-29

**Authors:** Mai Nishimura, Takehisa Nakanishi, Masako Ichishi, Yoshiaki Matsushima, Masatoshi Watanabe, Keiichi Yamanaka

**Affiliations:** 1Department of Dermatology, Mie University Graduate School of Medicine, 2-174 Edobashi, Tsu 514-8507, Mie, Japan; ishika-m@clin.medic.mie-u.ac.jp (M.N.); t-nakanishi@clin.medic.mie-u.ac.jp (T.N.); matsushima-y@clin.medic.mie-u.ac.jp (Y.M.); 2Inflammatory Skin Disease Research Center, Mie University Graduate School of Medicine, 2-174 Edobashi, Tsu 514-8507, Mie, Japan; 3Department of Oncologic Pathology, Mie University Graduate School of Medicine, 2-174 Edobashi, Tsu 514-8507, Mie, Japan; masako-i@doc.medic.mie-u.ac.jp (M.I.); mawata@doc.medic.mie-u.ac.jp (M.W.)

**Keywords:** inflammatory skin disease, atopic dermatitis, psoriasis, sepsis, lipopolysaccharide, high-fat diet, innate lymphoid cells, cytokine, Toll-like receptor 4, hypothermia

## Abstract

In recent years, attention has increasingly focused on various infectious diseases. Although some fatalities are directly attributed to the causative virus, many result from complications and reactive inflammation. Patients with comorbidities are at a higher risk of mortality. Refractory skin conditions such as atopic dermatitis, psoriasis, and epidermolysis bullosa, known for an elevated risk of sepsis, partly owe this to compromised surface barrier function. However, the detailed mechanisms underlying this phenomenon remain elusive. Conversely, although the detrimental effects of a high-fat diet on health, including the onset of metabolic syndrome, are widely recognized, the association between diet and susceptibility to sepsis has not been extensively explored. In this study, we examined the potential causes and pathogenesis of increased sepsis susceptibility in inflammatory skin diseases using a mouse dermatitis model: keratin 14-driven caspase-1 is overexpressed (KCASP1Tg) in mice on a high-fat diet. Our findings reveal that heightened mortality in the dermatitis mouse model is caused by the inflamed immune system due to the chronic inflammatory state of the local skin, and administration of LPS causes a rapid increase in inflammatory cytokine levels in the spleen. Intake of a high-fat diet exacerbates these cytokine levels. Interestingly, we also observed a reduced expression of Toll-like receptor 4 (TLR4) in monocytes from KCASP1Tg mice, potentially predisposing these animals to heightened infection risks and associated complications. Histological analysis showed a clear decrease in T and B cells in the spleen of KCASP1Tg mice fed a high-fat diet. Thickening of the alveolar wall, inflammatory cell infiltration, and alveolar hemorrhage were more prominent in the lungs of KCASP1Tg and KCASP1Tg with fat mice. We postulate that the chronic, non-infectious inflammation induces a negative feedback loop within the inflammatory cascade, and the suppressed expression of TLR4 renders the mice more susceptible to infections. Therefore, it is imperative for individuals with chronic skin inflammation to closely monitor disease progression upon infection and seek timely and appropriate treatment. Additionally, chronic inflammation of adipose tissue, induced by high-fat food intake, combined with dermatitis inflammation, may exacerbate infections, necessitating a review of dietary habits.

## 1. Introduction

As established at the 1991 Consensus Conference, sepsis was defined by the manifestation of systemic inflammatory response syndrome (SIRS) criteria in the presence of infection [1,2,3]. These criteria include changes in temperature, elevated pulse and respiratory rates, and abnormal white blood cell counts [1,2,3]. However, it is now widely accepted that sepsis can also be characterized as a severe and potentially fatal organ dysfunction resulting from an aberrant host response to infection [1,3,4]. It still affects 47 to 50 million people each year and kills at least 11 million—one death every 2.8 s. Although sepsis is associated with 20% of all deaths worldwide, its specific mechanism is not fully understood, and no specific treatment has been established (https://www.worldsepsisday.org/sepsis, accessed on 29 December 2023). The risk of mortality from sepsis varies depending on factors such as the patient’s age, gender, the infectious agent involved, the presence or absence of spleen, and basic immunity [5,6]. Moreover, pre-existing abnormalities in the circulatory system, the fragility of vascular endothelial cells, and dysfunctions in the cellular and metabolic systems markedly elevate mortality risk [1,6]. In addition, even if the patients escape death from sepsis, they may suffer from its sequelae for the rest of their lives: difficulty swallowing, poor mobility, joint and muscle pains, clouded thinking, difficulty sleeping, poor memory, difficulty concentrating, and fatigue [7,8,9].

Generally, patients with chronic inflammatory conditions, including refractory skin and connective tissue diseases, face an elevated risk of sepsis [10,11,12,13,14]. This heightened risk might arise from the effects of immunosuppressive drugs prescribed for these conditions [12,13,14,15] or from a compromised surface barrier function. Additionally, abnormalities in the vascular system, organ damage, and immune system anomalies linked with chronic inflammation could also play roles but have not been specifically tested. Unlike medical diseases such as hypertension and diabetes mellitus, patients with chronic dermatitis still lack awareness of the disease as a health risk threat, and many patients have neglected the disease for many years due to their busy schedules and financial burdens. Therefore, through this study, we aim to clarify the mortality risk of sepsis in patients with dermatitis, its pathogenesis and causes, and to educate patients about the seriousness of dermatitis treatment.

In the present study, we assess the risk of mortality in septic conditions, its pathogenesis, and its underlying causes using a mouse dermatitis model: KCASP1Tg mice, which overexpress keratin 14-driven caspase-1. In these mice, caspase-1 is overexpressed in the basal layer of keratinocytes, leading to the activation of interleukin (IL)-1β and IL-18. Beginning around the 8-week mark, these mice exhibit dermatitis, primarily on the face, which eventually extends across the body [16,17,18]. Skin scrapings and curettage further stimulate the production of cytokines such as IL-1α, IL-6, Tumor necrosis factor α (TNF-α), and IL-23. It is well-documented that if such a condition persists, it not only results in skin inflammation but also systemic inflammation, potentially leading to complications and a reduced lifespan [19,20].

In addition, dermatitis has recently been known to be exacerbated by the consumption of high-fat diets [21,22,23]. High-fat diet intake itself also induces obesity, which leads to adipose tissue inflammation and the release of adipocytokines, creating a chronic inflammatory state [24,25,26]. This, coupled with the inflammation of dermatitis, can lead to severe infections. Therefore, we investigated the risk of sepsis posed by dermatitis and the consumption of a high-fat diet.

## 2. Results

### 2.1. Survival Rate

Lipopolysaccharide (LPS) at a dose of 500 μg was administered intraperitoneally to wild-type (WT) mice and KCASP1Tg mice, as well as to each group of mice that were fed a high-fat diet (*N* = 5, each group). The cumulative survival rate is shown (Figure 1). In the WT and WT + fat mice groups, all mice survived even after 36 h. In contrast, a decrease in survival rate was observed in the KCASP1Tg and KCASP1Tg+ fat mouse groups. KCASP1Tg+ fat mice that were alive after 36 h exhibited low body temperatures (21.5 °C) compared to 29.4 °C in WT mice.

### 2.2. Cytokine-Producing Cells from the Spleen before and after Intraperitoneal Administration of LPS

We administered a small amount of LPS (10 μg) into the peritoneal cavity of a mouse, and we extracted the spleen after 12 h. Mononuclear cells were isolated, the monocyte population was gated morphologically, and innate lymphoid cells (ILCs) were gated, excluding the surface-specific marker-stained population. We then measured the cytokine expression in the monocytes and ILCs using flow cytometry. We present a graph summarizing the number of cells producing Type 1 cytokines TNFα and Interferon-γ (IFN-γ), Type 2 cytokines IL-4 and IL-13, and Type 3 cytokines IL-17A and IL-17F. In the monocyte population, IFN-γ-producing cells were increased in KCASP1Tg + fat mice after LPS administration. IL-4 levels increased in KCASP1Tg before and after LPS administration. IL-17A and F levels were unchanged (Figure 2A). For ILC, IFN-γ tended to increase in KCASP1Tg and KCASP1Tg + fat mice before LPS administration and significantly increased after LPS administration. TNF-α tended to increase in KCASP1Tg, and the difference was significant after LPS administration. IL-4 and IL-13 also increased in KCASP1Tg and KCASP1Tg + fat mice before LPS administration and significantly increased in KCASP1Tg + fat mice after LPS administration. IL-17F was significantly elevated in KCASP1Tg + fat mice before LPS administration and decreased after LPS administration compared to before. There was no apparent change in IL-17A (Figure 2B).

### 2.3. Expression of TLR4, the TLR4 Antagonist Lipopolysaccharide Binding Protein (LBP), Myeloid Differentiation Primary Response (Myd88), and TIR-Domain-Containing Adapter-Inducing Interferon-β (Ticom1), Downstream Signals of TLR4

We administered 10 μg of LPS into the mouse’s peritoneal cavity, extracted the spleen 12 h later, and purified monocytes. Messenger ribonucleic acid (mRNA) was isolated, and the expression of TLR4, a receptor that recognizes LPS, its antagonist LBP, and downstream signals of TLR4, Myd88, and Ticom1, were measured using reverse transcription polymerase chain reaction (RT-PCR). As a result, while the expression of TLR4 did not significantly change before and after LPS administration, a significant decrease in baseline expression of TLR4 was observed in both KCASP1Tg mice and KCASP1Tg + fat mice (Figure 3). As for the expression of LBP, an antagonist of TLR4, it was found to increase in mice on a high-fat diet and in KCASP1Tg mice, suggesting that it may naturally increase to suppress innate immunity. Myd88 and Ticam1 are downstream signals of TLR4 and were found to be decreased in mice on a high-fat diet and in KCASP1Tg mice, in proportion to the expression of TLR4.

### 2.4. Plasma Cytokine Levels

Twelve hours after administering 10 μg of LPS into the peritoneal cavity of mice, plasma was collected from each mouse, and various cytokines were measured (Figure 4). In WT mice, none of the cytokines, including IFN-γ, IL-2, IL-4, IL-5, IL-6, TNF-α, IL-9, IL-10, IL-13, IL-17A, IL-17F, and IL-22, were elevated. In contrast, in KCASP1Tg mice with LPS injection, while not statistically significant, there were tendencies for increases in IL-6, TNF-α, IL-17A, and IL-22. There was a significant increase in IL-10.

### 2.5. Histology of Organs

The tissues of each mouse were examined for the presence of organ damage. The pathology of the spleen is shown (Figure 5). Compared to WT, KCASP1Tg + fat mice showed a clear decrease in CD4-positive T cells, CD8-positive T cells, and CD20-positive B cells, while no obvious difference was observed in CD138-positive plasma cells. 

The pathological image of the spleen after 10 μg of LPS administration is shown (Figure 6). Compared to Figure 5, the pathology of the spleen showed a generally similar trend. KCASP1Tg + fat mice showed a decrease in CD8-positive T cells and CD20-positive B cells.

To examine other organ damage, hematoxylin and eosin (H&E)-stained images of the liver, kidney, and lung are shown (Figure 7). In the liver, none of the mice showed fatty degeneration, necrosis, or inflammatory cell infiltration. In the kidneys, no tubular epithelial degeneration or necrosis was observed. In the lungs, thickening of the alveolar wall, inflammatory cell infiltration, and alveolar hemorrhage were more prominent in KCASP1Tg and KCASP1Tg + fat mice compared to those in WT mice.

## 3. Discussion

KCASP1Tg mice have been regarded as a model of atopic dermatitis because their dermatitis characteristics and histological and behavioral profile meet 7 of 8 of Hanifin and Rajka’s diagnostic criteria for atopic dermatitis [16]. Established studies have shown that chronic inflammation of the skin is not only a local problem. It induces inflammation in organs throughout the body, causing cardiovascular and cerebrovascular disorders, systemic amyloidosis, and other organ damage [20,27,28,29,30,31,32,33,34,35]. Furthermore, it is known to induce inflammation in adipose tissue, with known effects such as weight loss due to adipose tissue atrophy, decreased heat production capacity, and increased production of inflammatory adipocytokines [23,36,37,38,39]. The prolonged consumption of a high-fat diet is also known to cause adipocyte hypertrophy and increased production of adipocytokines, leading to metabolic disorders such as insulin resistance, diabetes, and hepatitis [40]. In the current study, LPS was administered directly to these mice to create a septic state. LPS is a molecule that plays a central role in the onset and progression of sepsis associated with Gram-negative bacterial infections and has a variety of physiological activities, including toxicity, febrile response, and immunostimulatory effects [6,41,42,43]. In the late stages of bacterial infection, a phenomenon known as endotoxin shock can occur [6,42,44]. This is characterized by inflammatory cytokines (such as TNF-α and IFN-γ) produced by immune cells that increase blood coagulation capacity and vascular permeability, decrease blood pressure, and ultimately lead to peripheral circulatory failure [42].

In this study, we first examined the survival rates after LPS administration in WT mice, KCASP1Tg mice, and their respective mice by feeding a high-fat diet. The results showed that all five WT and WT + fat mice survived, but all KCASP1Tg mice died, and a decreased survival rate was also detected in KCASP1Tg + fat mice. In KCASP1Tg and KCASP1Tg + fat mice, marked hypothermia was observed. The cause of hypothermia may be due to ischemia of the thermoregulatory centers (hypothalamus and vestibular nuclei) caused by hypotension, decreased basal metabolism due to hypercytokinemia, heat dissipation due to skin disorders, decreased heat production due to emaciation, and also a combination of septic shock with vascular stenosis of dermatitis origin and peripheral circulation.

Next, we assessed cytokine production after 10 μg LPS administration. In the early stages of infection, the immune cells involved in inflammation include neutrophils, macrophages, monocytes, innate lymphoid cells (ILCs), dendritic cells, and NK cells. Recently, it is been pointed out that ILCs play a significant role in conditions like atopic dermatitis and psoriasis [45,46,47,48,49]. In this context, we focused on monocytes and ILCs to investigate factors that exacerbate sepsis in KCASP1Tg mice and high-fat diet mice. The results showed that type 1 cytokine IFN-γ and type 2 cytokine IL-4-producing monocytes were increased in KCASP1Tg mice, even without LPS administration. On the other hand, ILCs producing type 1 and type 2 cytokines tended to increase in KCASP1Tg and KCASP1Tg + fat mice. This may reflect local inflammation in the skin, i.e., activation of ILCs in the spleen in response to signals from epithelial cells or accelerated immune systems. In addition, ILCs producing IFN-γ were significantly increased in KCASP1Tg and KCASP1Tg + fat mice after LPS administration, indicating enhanced cytokine release coupled with existing inflammatory responses. Furthermore, IL-17F, a type 3 cytokine, was markedly increased in KCASP1Tg + fat mice even before LPS administration, and a trend of increase was also observed in WT + fat mice, where no other cytokine increases were observed. This is consistent with known reports that mice fed a high-fat diet tend to be obese [40] and that BMI positively correlates with IL-17F cytokine levels [50]. Obesity is associated with chronic inflammatory responses, including abnormal secretion of adipokines and activation of various inflammatory signaling pathways [51,52]. ILC3 is also known to promote Th17 cell differentiation in the spleen and has been reported to be activated by ILC3 upon IL-17 stimulation. From the above, it can be inferred that the basal production of IL-17F was increased and relatively decreased by the increased production of type 1 cytokines by LPS administration.

Furthermore, TLR4 expression was significantly reduced in KCASP1Tg and KCASP1Tg + fat mice. In addition, interacting with LPS, TLR4 also binds to DAMPs, which are released when cells are damaged or stressed, leading to non-infectious inflammation. In addition, saturated fatty acids are known non-microbial TLR4 agonists and activate the inflammatory cascade [53,54]. The HFD ratio used in this study is 60% calories derived from fat, 40% of which is composed of saturated fatty acids. In normal diets, this is only a few percent, clearly an overdose. Thus, the increase in DAMPs associated with chronic inflammation due to dermatitis and high-fat diet intake, as well as elevated saturated fatty acids in the blood, may lead to immune system exhaustion and negative feedback autoinhibition, resulting in decreased TLR4 expression. Suppression of TLR4 expression would be expected to result in the inability to effectively recognize LPS, and thus the immune response would not be properly initiated. As a result, defense against infection is impaired, and sepsis is more likely to occur, leading to increased morbidity and severity of infectious diseases.

Regarding cytokine concentrations in plasma, a significant increase in IL-10 was observed in the blood of KCASP1Tg mice, and IL-6 and TNF-α also tended to increase. In a cohort study examining cytokines in septic patients, the pattern of highest mortality risk was characterized by marked elevations in both the inflammatory cytokine IL-6 and the anti-inflammatory cytokine IL-10, consistent with the present findings [55]. This suggests that inflammatory and anti-inflammatory responses exist simultaneously and that a disturbance in homeostasis is present. In addition, there was an increasing trend in type 3 cytokines such as IL-17A and IL-22, suggesting an effect of skin inflammation, tissue repair, and infection on the production of anti-microbial peptides.

In this study, we employed H&E staining and immunohistological staining to assess the condition of various organs. The spleen of KCASP1Tg + fat mice, as revealed by H&E staining, displayed no significant follicular morphological changes or inflammatory cell infiltration. However, a notable decrease in CD4, CD8, and CD20-positive cells was observed, indicating an intrinsic defect in lymphocyte maturation. This aligns with existing literature that suggests the influence of amyloid deposition on such defects. Furthermore, while H&E staining revealed no apparent changes in the liver and kidney tissues, lung tissue analysis in both KCASP1Tg and KCASP1Tg + fat mice showed thickened alveolar walls, inflammatory cell infiltration, and alveolar hemorrhage. The absence of pulmonary and interstitial edema implies that the lung injury observed is likely due to hypercytokinemia, a result of dermatitis or LPS administration, rather than acute respiratory distress syndrome (ARDS).

In this study, we conclude that a prolonged inflammatory state in the skin sensitizes the immune system, with LPS administration causing a swift elevation in blood inflammatory cytokine levels. This response is further intensified by a high-fat diet. Furthermore, we observed a reduction in TLR4 expression in monocytes, a likely consequence of the negative feedback from non-infectious inflammation. Compounding these findings are impaired lymphocyte maturation in the spleen and lung tissue damage, which potentially underlie the increased susceptibility to infection and severe illness observed in our models. Interestingly, despite more severe pathologies in KCASP1Tg + fat mice compared to KCASP1Tg, there was an unexpected reversal in survival rates in our study. This discrepancy might be attributed to the differences in body size and physiology between mice and humans. Mice, with their smaller size, exhibit a higher surface area-to-volume ratio and lower heat retention, coupled with a higher basal metabolic rate. This necessitates greater energy expenditure to maintain body temperature, making them prone to hypothermia. The high-fat diet in mice may have enhanced subcutaneous fat accumulation, aiding in heat retention. However, this correlation may not directly apply to humans, where a high-fat diet has been linked to increased blood inflammatory cytokine levels and subsequent organ damage. Therefore, it is imperative that future clinical studies meticulously investigate this aspect to discern its implications for human health.

A limitation of this study is that it is being conducted in a single mouse model of dermatitis, and because of the intrinsic differences between humans and mice, the results of this study cannot be entirely substituted for humans. However, this study underscores that chronic dermatitis, if left untreated, extends beyond a mere cosmetic concern; it poses significant life-threatening risks by impacting internal organs and the immune system. The critical nature of treating this condition cannot be overstressed, and it is imperative to ensure that patients are thoroughly educated about these risks to appreciate the urgency of treatment. Moreover, this study highlights an additional concern: chronic inflammation of adipose tissue, often caused by high-fat diets, exacerbates the severity of infections when combined with dermatitis inflammation. This finding suggests the necessity of not only addressing the dermatological condition but also reviewing and potentially modifying dietary habits to mitigate these compounded health risks.

## 4. Materials and Methods

### 4.1. Animals

Female transgenic mice (16 weeks old) in which keratinocytes specifically overexpress the human caspase-1 gene under the keratin 14 promoter (KCASP1Tg) [16] were used as dermatitis models, and C57BL/6N littermates (wild type: WT) mice were used as controls. The Mie University Board Committee approved the experimental protocol for Animal Care and Use (#22-39-6-1, approval date: 2 June 2023). All mice were housed under specific environmental controls (temperature: 21 ± 2 °C, humidity: 60%, light cycle: 12/12 h) and allowed free access to food and water. In the high-fat diet mice, a high-fat feed comprising 60% fat calories (High Fat Diet60, Oriental Yeast Co., Ltd., Tokyo, Japan) was administered for two months [56,57,58]. To simulate the state of sepsis, lipopolysaccharide LPS (Sigma-Aldrich, Co., St. Louis, MO, USA) adjusted to various concentrations using Phosphate-Buffered Saline (PBS, Nacalai Tesque, Kyoto, Japan) was directly administered into the mice’s abdominal cavity [59,60].

### 4.2. Tissue Sampling

All mice were subjected to euthanasia with CO_2_ or pentobarbital, and various organs were sampled. Additionally, whole blood was sampled through a cardiac puncture and placed in untreated tubes containing heparin (Mochida Pharmaceutical Co., Ltd., Tokyo, Japan). The samples were centrifuged at 1500 rpm for 10 min at 4 °C. The obtained serum was stored at −80 °C until testing.

### 4.3. Flow Cytometry Analysis

The number of ILCs producing various cytokines in the spleen was measured using flow cytometry (FCM). Analysis was performed on 5–8 animals per group. The spleen was crushed in a petri dish, and the cells were filtered through a mesh and incubated with ACK Lysing Buffer (Thermo Fisher Scientific, Waltham, MA, USA) to lyse the red blood cells. Mononuclear cells were isolated and purified by density gradient centrifugation. The LIVE/DEAD™ Fixable Aqua Dead Cell Stain Kit (Thermo Fisher Scientific) was used to exclude apoptotic and necrotic cells. The cultured mononuclear cells were stained with surface antibodies: CD45R-PerCp-Cy5.5, CD3-PerCp-Cy5.5, CD4-PerCp-Cy5.5, FceRI-PerCp-Cy5.5, CD8a-PerCp-Cy5.5, Gr-1-PerCp-Cy5.5, Siglec-F-PerCp-Cy5.5 (BD Biosciences, Franklin Lakes, NJ, USA) in cell surface staining buffer containing 0.1 M phosphate-buffered saline and 2% FCS (Biowest, Nuaillé, France) [61,62], and then stained with IFNγ-Brilliant Violet 605, TNFα-APC, IL-4-Brilliant Violet 421, IL-13-FITC, IL-17A-APC-Cy7, and IL-17F-PE antibodies (BD Biosciences). The expression patterns of inflammatory cytokines were analyzed using a BD Lyric flow cytometer (BD Biosciences), and data were analyzed using FlowJo software (v10.9.0) (Tree Star Inc., Ashland, OR, USA).

### 4.4. Real-Time Polymerase Chain Reaction (Real-Time PCR)

The spleen was collected, mononuclear cells were isolated, and then purified by density gradient centrifugation. Monocytes were isolated using CD11b MicroBeads (Miltenyi Biotec, Tokyo, Japan). Total RNA was extracted, and the RNA concentration was measured using a NanoDrop Lite spectrophotometer (Thermo Fisher Scientific, Worsham, MA, USA). Approximately 1 µg of total RNA was converted to cDNA using a High-Capacity RNA-to-cDNA Kit (Applied Biosystems, Foster City, CA, USA). The TaqMan Universal PCR Master Mix II with UNG (Applied Biosystems) was used to measure the mRNA expression of TLR4 (Mm00445273_m1, Mm00445274_m1), a receptor that recognizes LPS, its antagonist LBP (Mm00493139_m1), and downstream signals of TLR4, Myd88 (Mm00440338_m1), and Ticom1 (Mm00844508_s1). Glyceraldehyde-3-phosphate dehydrogenase (GAPDH, Mm99999915_g1) was used as an internal control. All probes were purchased from Thermo Fisher Scientific, and the amplification was performed in a LightCycler 96 System (Roche Diagnostics, Indianapolis, IN, USA). The cycling parameters were as follows: 50 °C for 120 s, 95 °C for 600 s, followed by 50 cycles of amplification at 95 °C for 15 s and 60 °C for 60 s.

### 4.5. Measurement of Plasma Cytokines

Plasma cytokine concentration was measured using the LEGENDplex MU Th Cytokine Panel (12-plex) w/FP V03 (BioLegend, San Diego, CA, USA) and flow cytometry BD Accuri TM C6 (BD Biosciences).

### 4.6. Histological Analysis

Each organ was collected and fixed in a 10% formalin neutral buffer solution (Wako, Osaka, Japan) for 12 h, dehydrated in an ethanol series (Wako), embedded in paraffin, cut into 6 µm sections, and stained with hematoxylin and eosin (H&E)**.** In addition, to identify the specific type of infiltrating cells in the spleen, CD4, CD8, CD20, and CD138 immunohistological staining was performed in the spleen. Positive cells were counted in five random fields of view at ×100 magnification for each sample. The percentage of cells stained by immunostaining in the total field of view was measured using NIH images.

### 4.7. Statistical Analysis

A statistical analysis was performed using PRISM software version 9 (GraphPad, San Diego, CA, USA). The Kruskal–Wallis test, followed by multiple comparisons, was used to compare groups. *p* values < 0.05 were considered indicative of statistically significant differences: a, *p* < 0.05; b, *p* < 0.01; c, *p* < 0.001; and d, *p* < 0.0001.

## Figures and Tables

**Figure 1 ijms-25-00478-f001:**
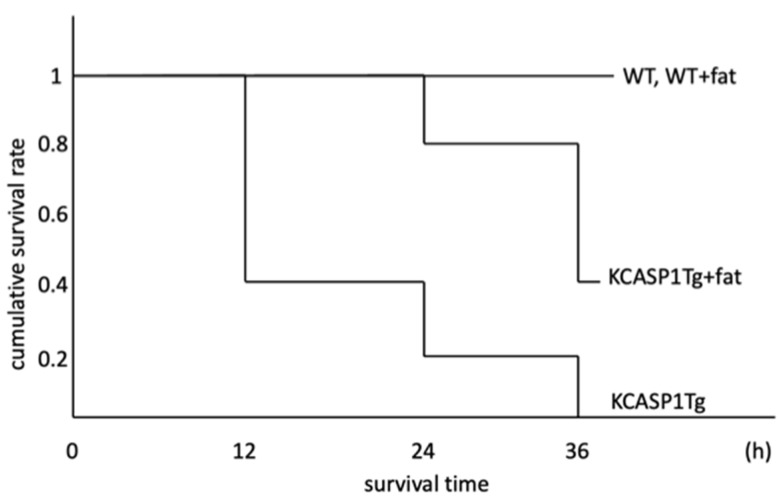
Survival rate after LPS administration. The cumulative survival rate after intraperitoneal administration of 500 μg LPS is shown. In the WT and WT + fat mice groups, all mice survived through the course of the observation. A decrease in survival rate was observed in both the KCASP1Tg and KCASP1Tg + fat mice groups.

**Figure 2 ijms-25-00478-f002:**
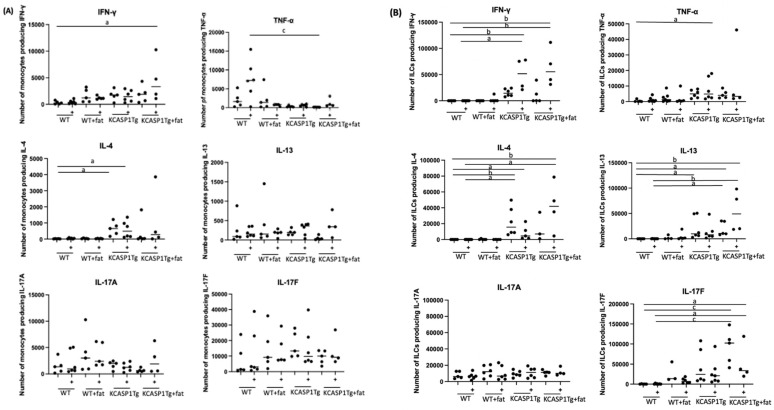
Comparison of cytokine-producing monocytes and ILC in the spleen. (**A**) The number of cytokine-producing cells in the monocytes. (+) means after 10 μg of LPS administration. In the monocyte population, there was an increasing trend in IFN-γ and IL-4 in both KCASP1Tg mice and KCASP1Tg + fat mice, and there was a significant increase in IFN-γ production after LPS administration. IL-17A and IL-17F were unchanged. (**B**) The number of cytokine-producing cells in the ILCs. (+) means after 10 μg of LPS administration. In ILCs, compared to WT, there was a tended increase in type 1 cytokines (TNF-α, IFN-γ) and type 2 cytokines (IL-4, IL-13) in both KCASP1Tg mice and KCASP1Tg + fat mice and further increase in IFN-γ, IL-4, and IL-13 cytokines after LPS administration. IL-17F was significantly elevated in KCASP1Tg + fat mice before LPS administration and decreased after LPS administration (**B**). A significant difference was indicated based on a one-way ANOVA and Tukey’s multiple comparison tests (a, *p* < 0.05; b, *p* < 0.01; c, *p* < 0.001).

**Figure 3 ijms-25-00478-f003:**
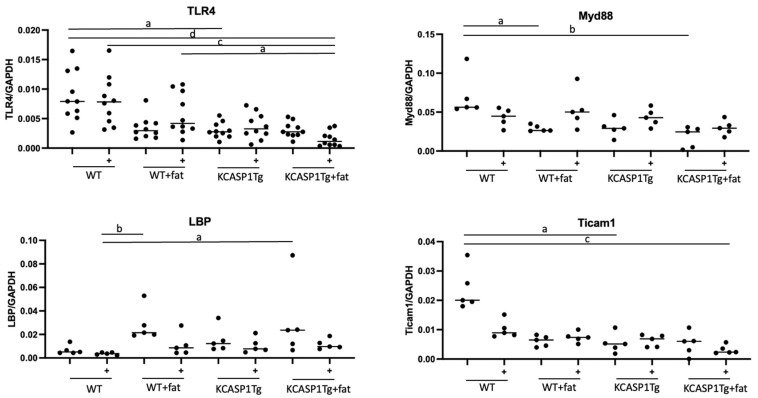
Expression of TLR4, LBP, Myd88, and Ticom1 in monocytes of the spleen. Expression of TLR4, LBP, Myd88, and Ticom1 in splenic monocytes before and after intraperitoneal administration of 10 μg of LPS. (+) is after LPS administration. A significant decrease in baseline expression of TLR4 was observed in both KCASP1Tg mice and KCASP1Tg + fat mice. LBP was increased in mice on a high-fat diet and in KCASP1Tg mice. Myd88 and Ticam1 are downstream signals of TLR4 and were decreased in mice on a high-fat diet and in KCASP1Tg mice. Significant differences were detected based on a one-way ANOVA and Tukey’s multiple comparison test (a, *p* < 0.05; b, *p* < 0.01; c, *p* < 0.001; d, *p* < 0.0001).

**Figure 4 ijms-25-00478-f004:**
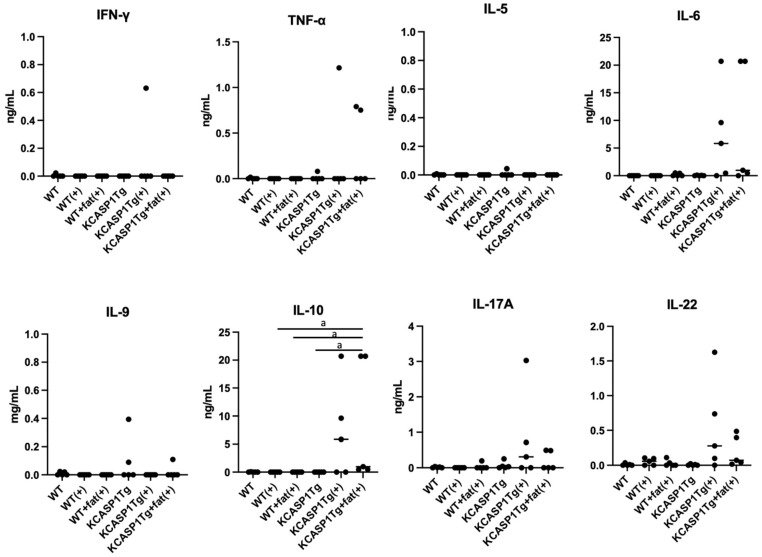
The plasma cytokine levels. Plasma cytokine levels 12 h after intraperitoneal administration of 10 μg LPS or without LPS injection were measured. (+) is with the LPS administration. None of the cytokines were elevated in WT and WT + fat mice; in contrast, in KCASP1Tg and KCASP1Tg + fat mice after LPS injection, there was a significant increase in IL-10, and while not statistically significant, there were tendencies of increases in IL-6, TNF-α, IL-17A, and IL-22. There were significant differences in IL-10 concentration based on a one-way ANOVA and Tukey’s multiple comparison test (a, *p* < 0.05).

**Figure 5 ijms-25-00478-f005:**
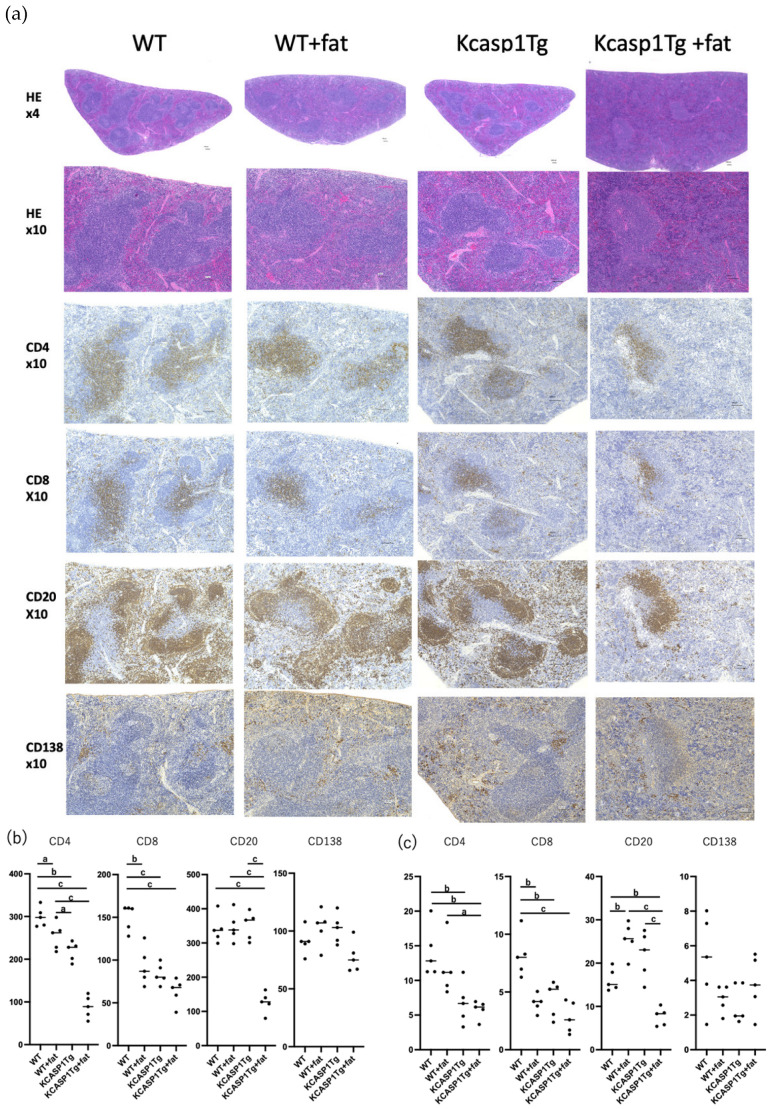
H&E-stained image of the spleen from 16-week-old mice. Although there was no obvious follicular dysplasia, a decrease in CD4-positive, CD8-positive T cells, and CD20-positive B cells was observed in KCASP1Tg + fat. No apparent differences were observed in CD138-positive plasma cells. *N* = 5. The representative pictures are shown in (**a**). Positive cells were counted in 5 random fields of view at ×100 magnification for each sample, with a significant decrease in staining for CD4-positive, CD8-positive T cells, and CD20-positive B cells in the KCASP1Tg + fat group (**b**). Similarly, statistical calculation of the percentage of cells stained by immunostaining in the total field of view using NIH images showed that CD4-positive, CD8-positive, and CD20-positive cells were predominantly lower in the KCASP1Tg + fat group (**c**). A significant difference was indicated based on a one-way ANOVA and Tukey’s multiple comparison tests (a, *p* < 0.05; b, *p* < 0.01; c, *p* < 0.001).

**Figure 6 ijms-25-00478-f006:**
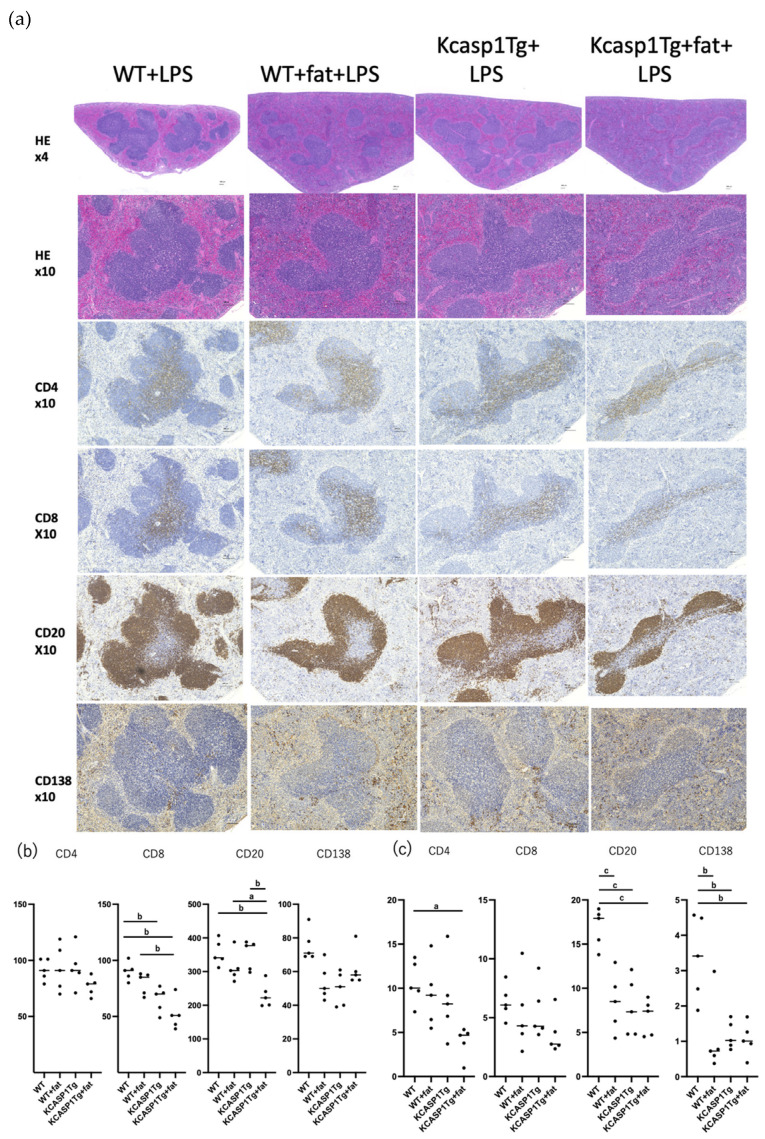
H&E stained the image of the spleen after 10 μg of LPS administration. There were no apparent changes in appearance compared to the spleen without LPS administration. KCASP1Tg + fat mice showed a decreased tendency in CD4-positive and CD8-positive T cells and CD20-positive B cells. *N* = 5. The representative pictures are shown in (**a**). Positive cells were counted in 5 random fields of view at ×100 magnification for each sample, with a significant decrease in staining for CD8-positive T cells and CD20-positive B cells in the KCASP1Tg + fat group (**b**). The percentage of cells stained by immunostaining in the total field of view using NIH images showed that CD4-positive and CD20-positive cells were statistically lower in the KCASP1Tg + fat group (**c**). A significant difference was indicated based on a one-way ANOVA and Tukey’s multiple comparison tests (a, *p* < 0.05; b, *p* < 0.01; c, *p* < 0.001).

**Figure 7 ijms-25-00478-f007:**
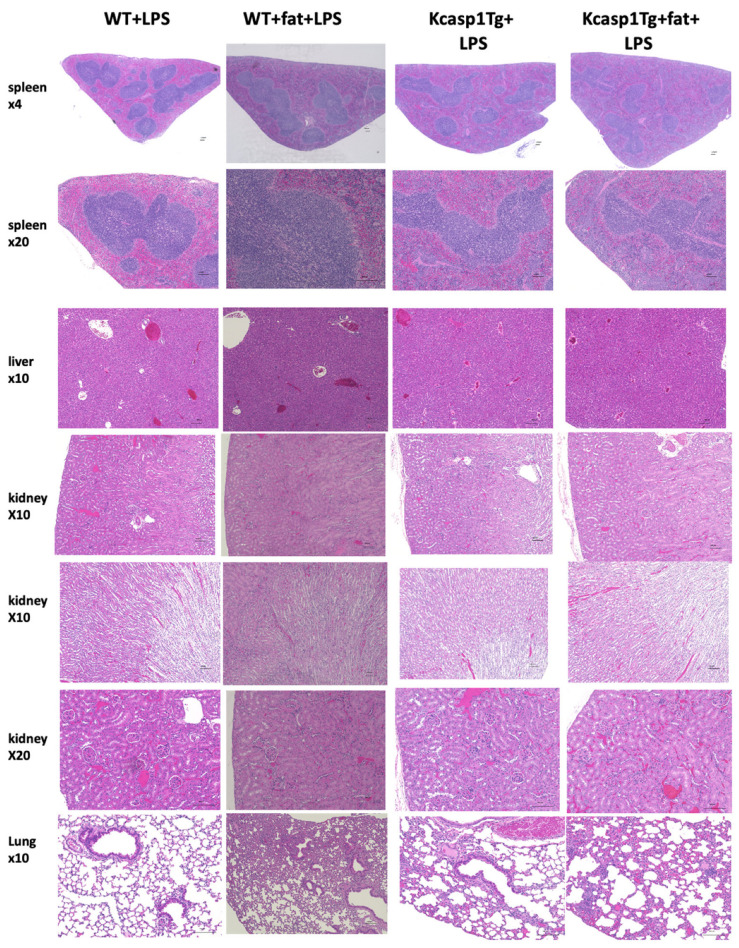
H&E-stained images of the spleen, liver, kidney, and lung after LPS administration. No obvious changes were observed in the spleen, liver, or kidneys. In the lungs of KCASP1Tg and KCASP1Tg + fat mice, alveolar wall thickening, inflammatory cell infiltration, and alveolar hemorrhage were observed. *N* = 5. Representative pictures were shown.

## Data Availability

The data presented in this study are available on request from the corresponding author.

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
