# Peer review of "Increased Mortality Risk at Septic Condition in Inflammatory Skin Disorders and the Effect of High-Fat Diet Consumption"

_ijms, 2023, doi:10.3390/ijms25010478_

Round 1

Reviewer 1 Report

Comments and Suggestions for Authors

The authors suggested that the persistent, non-infectious inflammation creates a negative feedback loop in the inflammatory cascade, making the mice more vulnerable to infections as a result of TLR4's decreased expression.

1.      Introduction may be improved adding new information in order to provide an adequate state-of-the-art.

2.      Discussion may include future research studies secondary to the current findings of this study.

3.      Provide the significance of study at the end of discussion.

4.      Figure ligands of figures such as 5, 6, and 7 should be more informative.

5.      In the materials and methods section, some methods lack references.

6.      In conclusion, the authors should highlight their major findings from this study. The authors should include the limitations of this study and future prospects.

7.      The influence of the consumption of dietary fats on body weight and obesity in relation with inflammation could be discussed.

8.      Name the primer designation tool in method section.

Comments on the Quality of English Language

I suggest checking for some minor spelling and grammar mistakes. 

Author Response

Responses to the comments of Reviewer #1

The authors suggested that the persistent, non-infectious inflammation creates a negative feedback loop in the inflammatory cascade, making the mice more vulnerable to infections as a result of TLR4's decreased expression.

  1. Introduction may be improved adding new information in order to provide an adequate state-of-the-art.

Response: Thank you for your suggestion. This has been corrected.

  1. Discussion may include future research studies secondary to the current findings of this study.

Response: We appreciate your suggestion. We have added our future research studies at the end of the discussion.

  1. Provide the significance of study at the end of discussion.

Response: Thank you for your advice. We have added.

  1. Figure ligands of figures such as 5, 6, and 7 should be more informative.

Response: This has been supplemented.

  1. In the materials and methods section, some methods lack references.

Response: Thank you for your suggestion. We have revised it.

  1. In conclusion, the authors should highlight their major findings from this study. The authors should include the limitations of this study and future prospects.

Response: Thank you for your suggestion. We have revised the text.

  1. The influence of the consumption of dietary fats on body weight and obesity in relation with inflammation could be discussed.

Response: Thank you for your great suggestion. We have added a discussion on the relationship between dietary fat obesity and inflammation.

  1. Name the primer designation tool in method section.

Response: Thank you for your suggestion. We have changed.

Comments on the Quality of English Language: I suggest checking for some minor spelling and grammar mistakes. 

Response: Thank you for your suggestion. The text was revised and corrected by a native English-speaking scientist. Again, we appreciate your comments.

Reviewer 2 Report

Comments and Suggestions for Authors

In this paper, Nishimura et al. investigated the effects of the high-fat diet on inflammatory skin diseases using KCASP1Tg mice. The quality of this paper needs to be improved, and currently, there are a lot of issues with this study, and data need to be further interpreted. Given the current state of this paper, I regretfully recommend rejection. Here are some comments on this study:

1.       Abstracts need to be refined.

2.       Lines 45-46 “…criteria in the presence of infection.” require a reference.

3.       The introduction needs to be revised and refined. The logic of the current introduction was confusing. The background of the study, the shortcomings of the current studies, and the significance of this study should be clearly presented, in particular by stating the reasons why studying high-fat diet is important.

4.       Line 83 “body temperatures” could the authors provide the specific body temperature data?

5.       From Figure 1, we could find that HFD does not affect the survival rate of WT mice, rather it improved the survival rate of KCASP1Tg mice. Does this mean that HFD has a positive effect?

6.       In section 2.1, the authors used 500 μg LPS, but 10 μg LPS was used in later experiments. From Figure 1, not all mice in KCASP1Tg group died at 12 h. Why not do the experiment at 500 μg LPS?

7.       Significance labeling is confusing and it is recommended that letters (a,b,c…) be used to label significant differences.

8.       The content of the legend for Figure 2 was confusing and the authors should to revise it. What is more, we found that there was great variability in the Figure 2 data, especially in KCASP1Tg and KCASP1Tg+hfd groups. Under LPS treatment, the TNF-a levels did not increase in KCASP1Tg and KCASP1Tg+hfd groups, could the authors give any explanation?

9.       Figure 4 appears to have a major issue. Since IL-13 and IL-17 were all 0, it is recommended that they be removed. From Figure 4, neither LPS nor HFD appeared to increase inflammatory factors, can this author give some explanation for this?

10.    Line 169 “hematoxylin and eosin (HE)” should be “H&E”.

11.    It is difficult to compare the results from the images. Could authors provide the quantitative results of histology?

12.    Lines 221-224 “During this 221 period, the immune cells involved in inflammation include neutrophils, macrophages, 222 monocytes, innate lymphoid cells (ILCs), dendritic cells, and NK cells” was confusing, please revise it.

13.    Line 281 “. Although these pathologies 281 are more severe in KCASP1Tg+fat mice than in KCASP1Tg,” Does this mean HFD has a positive effect?

Author Response

Responses to the comments of Reviewer #2

In this paper, Nishimura et al. investigated the effects of the high-fat diet on inflammatory skin diseases using KCASP1Tg mice. The quality of this paper needs to be improved, and currently, there are a lot of issues with this study, and data need to be further interpreted. Given the current state of this paper, I regretfully recommend rejection. Here are some comments on this study:

  1. Abstracts need to be refined.

Response: Thank you for your suggestion. This has been corrected.

  1. Lines 45-46 “…criteria in the presence of infection.” require a reference.

Response: We appreciate your suggestion. We have added the references.

  1. The introduction needs to be revised and refined. The logic of the current introduction was confusing. The background of the study, the shortcomings of the current studies, and the significance of this study should be clearly presented, in particular by stating the reasons why studying high-fat diet is important.

Response: Thank you for your advice. We have added information and made revisions to clarify the background and objectives of this study. Thank you for your accurate advice.

  1. Line 83 “body temperatures” could the authors provide the specific body temperature data?

Response: This has been supplemented. We have appended specific temperature data.

  1. From Figure 1, we could find that HFD does not affect the survival rate of WT mice, rather it improved the survival rate of KCASP1Tg mice. Does this mean that HFD has a positive effect?

Response: Thank you for pointing this out. The apparent survival rate of obese KCASP1Tg mice on HFD is higher than that of obese KCASP1Tg mice because the mice are much smaller than humans, have a higher surface-to-volume ratio and lower heat retention capacity, and have a higher basal metabolic rate and thus have a greater energy requirement to maintain body temperature. The higher basal metabolic rate of the mouse is assumed to result in a greater energy requirement to maintain body temperature. However, since there is a lack of evidence to determine that ingestion of HFD has a positive effect because of the exacerbation of cytokines in the blood, we believe that this is an issue for future research. This has been supplemented in the text.

  1.  In section 2.1, the authors used 500 μg LPS, but 10 μg LPS was used in later experiments. From Figure 1, not all mice in KCASP1Tg group died at 12 h. Why not do the experiment at 500 μg LPS?

Response: Thank you for your suggestion. In Figure 1, a high dose of 500 μg of LPS was administered to assess mortality risk. However, we determined that administering 500 μg cannot capture the state of the immune response in the early stages of infection because it would lead to an early transition to a severe condition. The immune neck stone of dermatitis is the ILC, and ILCs are involved in the immune response from the early stages of infection. We believe that examining the cytokine profile in the early stages will help us understand the pathophysiology, i.e., what kind of immune abnormalities are occurring, and will also lead to risk assessment. Therefore, we experimented with a dose of 10 μg, which is the dose that does not kill mice.

  1. Significance labeling is confusing and it is recommended that letters (a,b,c…) be used to label significant differences.

Response: Thank you for your suggestion. We have changed all the significantly different labels.

  1. The content of the legend for Figure 2 was confusing and the authors should to revise it. What is more, we found that there was great variability in the Figure 2 data, especially in KCASP1Tg and KCASP1Tg+hfd groups. Under LPS treatment, the TNF-a levels did not increase in KCASP1Tg and KCASP1Tg+hfd groups, could the authors give any explanation?

Response: Thank you for your suggestion. Legend in Figure 2 has been modified: TNF-α is associated with chronic inflammatory conditions, and its production is known to be increased in both dermatitis and obesity; the elevation of plasma IL-10 in KCASP1Tg and KCASP1Tg+fat mice suggests that the increased IL-10, an immunosuppressive factor may inhibit the production of TNF-α. In addition, the production of TNF-α is concentrated in the skin and adipose tissue, which may be a reason for the decreased production of TNF-α in the spleen, but this is a subject for future research. Thank you very much for your very useful suggestions.

  1. Figure 4 appears to have a major issue. Since IL-13 and IL-17 were all 0, it is recommended that they be removed. From Figure 4, neither LPS nor HFD appeared to increase inflammatory factors, can this author give some explanation for this?

Response: Thank you for your suggestion. It is possible that the cytokine levels in the blood have not been significantly affected because the condition is captured in the very early stages of infection or that the immune system is abnormal in the direction of controlling the inflammatory response due to chronic inflammation, resulting in a marked increase in IL-10, an anti-inflammatory cytokine, but not yet an increase in inflammatory cytokines.

  1. Line 169 “hematoxylin and eosin (HE)” should be “H&E”.

Response: Thank you for your suggestion. We have changed it.

  1. It is difficult to compare the results from the images. Could authors provide the quantitative results of histology?

Response: Thank you for your suggestion. We have supplemented the quantitative data in the revised version.

  1. Lines 221-224 “During this 221 period, the immune cells involved in inflammation include neutrophils, macrophages, 222 monocytes, innate lymphoid cells (ILCs), dendritic cells, and NK cells” was confusing, please revise it.

Response: Thank you for your suggestion. We have changed our wording to say early stages of infection. Thank you.

  1. Line 281 “. Although these pathologies are more severe in KCASP1Tg+fat mice than in KCASP1Tg,” Does this mean HFD has a positive effect?

Response: Thank you for your suggestion. In this mouse experiment, the survival rate was higher in KCASP1Tg+fat mice compared to that in KCASP1Tg mice. Since humans and mice have different body sizes, different heat production requirements, and different heat retention capacities, we consider the possibility that obesity may have had a positive effect on body temperature retention in mice. However, since cytokine exacerbation, decreased TLR4 expression, and organ damage were exacerbated by HFD intake, we do not believe that the same effect may be observed in humans. This is a subject for future research and should be carefully investigated through clinical studies in humans.

Round 2

Reviewer 2 Report

Comments and Suggestions for Authors

Many thanks to the authors for the reply and modification.

For significance labeling, it is recommended that authors refer to this site: https://github.com/vicruiser/tukey_test_plot The current way of labeling is quite confusing.

For Figures such as Figure 2 a IL-13 and IL-17, it is appropriate to reduce the scale of the y-axis.

Author Response

Responses to the comments of Reviewer #2

Many thanks to the authors for the reply and modification.

Comment 1: For significance labeling, it is recommended that authors refer to this site: https://github.com/vicruiser/tukey_test_plot The current way of labeling is quite confusing.

Response: Thank you for your suggestions regarding statistical analysis. In our study, we employed the Kruskal-Wallis test and multiple comparison methods for between-group comparisons as nonparametric tests. This approach was chosen to compare three or more independent samples based on our determination that the groups were independent of each other and did not follow a normal distribution. While we conducted our analysis using PRISM software, the interpretation of the results is complicated due to the numerous significant differences observed within each group. We have also consulted with our statistician.

Comment 2: For Figures such as Figure 2 a IL-13 and IL-17, it is appropriate to reduce the scale of the y-axis.

Response: Thank you for your suggestion. This has been modified.
